# Effect of Wet Grinding Concrete Slurry Waste on Hydration and Hardening Properties of Cement: Micro-Nano-Scale Modification

**DOI:** 10.3390/ma17123010

**Published:** 2024-06-19

**Authors:** Guishan Liu, Hao Sun, Yongbo Huang, Peng Du

**Affiliations:** 1Provincial Key Laboratory of Preparation and Measurement of Building Materials, University of Jinan, Jinan 250022, China; 202121100331@stu.ujn.edu.cn (G.L.); mse_huangyb@ujn.edu.cn (Y.H.); 2Shandong Luqiao Building Materials Co., Ltd., Jinan 250000, China; 13853172340@163.com

**Keywords:** concrete slurry waste, wet grinding, hydration, compressive and flexural strength

## Abstract

The concrete slurry waste (CSW) produced by concrete mixing plants is a type of hazardous waste that is difficult to handle. To better recycle the CSW separated from the aggregates, this study uses a variety of wet-grinding processes to refine the solid in it, replaces some of the cement with the solid particles in wet grinding concrete slurry waste (WCSW), and investigates the properties of WCSW and its effect on the hydration and hardening properties of cement. The results show that a suitable wet-grinding process can ensure that the particle size in WCSW is less than 10 μm, the particle morphology is more flat, and the degree of hydration is higher. The WCSW particles can promote early cement hydration; after adding WCSW, the heat release peak of cement hydration appears earlier and more early hydration products are produced, and with the increase in the substitution amount, the promoting effect on early cement hydration will be more significant. The WCSW particles have a great effect on improving the strength of mortar, especially in the early stage. At 1 d, when the substitution amount is 7.5 wt.%, the compressive and flexural strength is increased by 43.67% and 45.04%; this is related to the filling of matrix pores and the improvement of the interface transition zone by micro- and nanoparticles. After the wet grinding of CSW, fine WCSW particles are obtained, which can improve the performance of cement-based materials by replacing cement.

## 1. Introduction

In modern construction engineering, higher requirements are placed on the quality and construction speed of concrete, resulting in a sharp increase in the demand for high-quality ready-mixed concrete [1,2]. In the concrete mixing plants, the centralized production mode is conducive to ensuring the productivity and production quality of concrete; hence, the mixing plant undertakes a lot of concrete production work. After the concrete is produced and transported to the designated point for use, the mixing plant will flush the residual concrete from the tank trucks of transportation, equipment and site of production with water and generate a large amount of waste, after which the aggregates will be separated for recycling, at the same time, the remaining concrete slurry waste (CSW) will be collected in the sedimentation tank for treatment [3,4,5,6]. There are hydrated and unhydrated cement and mineral admixtures in CSW, which are alkaline and will cause serious harm to the surrounding environment after discharge [7,8,9].

Hardened CSW can be crushed into recycled aggregates to produce concrete. However, the many drawbacks of CSW aggregate, such as its low density, high porosity, and high water absorption [10,11,12], can seriously affect the properties of concrete [13]. In addition to being used as aggregate, dry CSW powder can be obtained by grinding treatment, which has similar physical properties to fly ash, slag and other admixtures so that they can replace some cementitious materials [14].

The wet-grinding process can also effectively grind large particle materials and usually has a better grinding efficiency and effect [15,16,17]. Micro-nanoparticles can be obtained by wet grinding CSW, and these fine wet grinding concrete slurry waste (WCSW) particles can be used as excellent auxiliary cementitious materials [18,19]. In the complex hydration and hardening process of cement-based materials, the addition of auxiliary cementitious materials can usually play a modifying role; in particular, the addition of fine auxiliary cementitious materials is more beneficial for optimizing the properties of cement-based materials [20,21,22].

In previous studies, CSW was mostly recycled after drying and crushing, while there has been less research on WCSW; therefore, in order to reduce the harm of CSW to the environment, further research is needed on the treatment process and properties of WCSW as well as its role in cement-based materials.

This study utilizes various wet-grinding processes to refine the solids in CSW and tests the size of the particles in WCSW through particle size distribution (PSD) analysis. The group with the best treatment effect, that is, the smallest particle size, was selected, and scanning electron microscope (SEM) analysis, X-ray diffraction (XRD) analysis and thermogravimetric (TG) analysis characterized the morphology of solid particles and main mineral components. Meanwhile, replacing cement with solid particles in WCSW and their effect on the hydration and hardening properties of cement are studied by hydration heat, XRD, TG, SEM and strength test methods.

## 2. Materials and Methods

### 2.1. Raw Materials

We obtained PO 42.5 cement from the Wohushan Cement Plant in Jinan and fly ash (FA) from the Xinyuan Power Plant in Liaocheng. In addition, we used ISO standard sand and polycarboxylic acid superplasticizer with a solid content of about 10 wt.% and a recommended water reduction rate of about 20%. The main chemical components of cement and FA obtained from X-ray fluorescence spectroscopy (XRF) are shown in Table 1.

### 2.2. Treatment and Characterization of CSW

Referring to the proportion of C45 strength grade concrete in the mixing plant, cement, FA, water and superplasticizer were mixed in a ratio of 42:8:200:1.25 to form CSW and then aged at room temperature for 12 h. After aging, the CSW at the bottom will continuously bond and harden into large-sized materials, and different wet-grinding schemes were used to refine it into smaller-sized materials.

CSW, grinding medium and water were mixed in a 1 L ball mill tank, where the ratio of solid, grinding medium and water is 1:3:5. Then, different wet-grinding schemes were designed to refine the CSW by changing the grinding medium, the grinding speed and the grinding time, as shown in Table 2.

The particle size distribution of solid particles in WCSW obtained through wet grinding are measured using a laser particle size analyzer, SEM was used to view the morphology of the solid particles, and XRD and TG analysis were used to determine the main mineral phases present in the solid particles.

### 2.3. Mix Design and Sample Preparation

Mortar were prepared by replacing cement with solid components in WCSW. The proportion of solid components replacing cement in WCSW was designed to be 0, 2.5 wt.%, 5 wt.%, 7.5 wt.% and 10 wt.%. The basic ratio of mortar is shown in Table 3.

For sample preparation, first, we mixed the WCSW, water, and superplasticizer for ultrasonic dispersion; then, we mixed the cement and FA with them and stirred for 1 min in a planetary mortar mixer. After, we added standard sand and continued stirring for 3 min. The mortar samples were formed in the iron mold of 40 × 40 × 160 mm, and the bubbles were discharged through vibration. After forming, the samples were placed in the curing chamber (temperature 20 ± 1 °C, humidity above 95%) for curing. After 1 d, the mold was removed, and the samples continued to be cured in the curing chamber for different ages.

Isopropanol can replace water in the sample and cause minimal damage to the sample structure; thus, it was used to terminate the hydration of the sample [23]. The aged samples were broken into thin pieces with a thickness of less than 3 mm and soaked in isopropanol, and the new isopropanol was replaced at 12, 36, and 72 h.

### 2.4. Test Method

#### 2.4.1. Particle Size Distribution (PSD)

The PSD of solids in WCSW was measured by a laser particle size analyzer (LS13320, Beckman Kurt, Boulevard Brea, CA, USA) with a measurement range of 0.04–2000 μm, and anhydrous ethanol was used as the dispersion medium. Due to the small particle size of WCSW particles, they were prone to cluster phenomena. Therefore, before detection, WCSW was mixed with superplasticizer and then assisted with ultrasonic treatment for dispersion.

#### 2.4.2. Hydration Heat

An 8-channel constant temperature calorimeter (TAMAIR, American Waters Corporatron, Milford MA, USA) was used to monitor the hydration thermal evolution of the slurry. In this test, the solids (cement and solid components in WCSW) of 4 g, water (water and water in WCSW) of 1.1 g, superplasticizer of 0.1 g, using an analytical balance to weigh accurately, were placed in a plastic cylinder bottle. We waited for the baseline to stabilize, performed manual stirring for 2 min, put the sample in the instrument and started recording. The whole test was kept at 20 °C, and the hydration heat release rate curve within 72 h and the hydration cumulative heat release curve within 0.5–72 h were recorded.

#### 2.4.3. XRD Analysis

The hydrated and dried samples were ground into a fine powder that could pass through the 200 mesh sieve and then tested with XRD (D8Advance, Bruker AXS, Karlsruhe, Germany), CuKα radiation (λ = 1.5418 A), scanning angles from 2θ = 5° to 2θ = 80°, and a step size of 0.12°. In quantitative analysis, αAl_2_O_3_ (0.4 μm) was used as the internal standard, and its mass accounted for 20%. The results were analyzed using TOPAS 4.2 software to determine the mass fraction of each phase [24]. The hydration degree of C_3_S minerals was characterized by the ratio of C_3_S minerals undergoing a hydration reaction to the total C_3_S minerals.

#### 2.4.4. SEM Analysis

After the termination of hydration, the samples were sequentially dried, sprayed with gold, and vacuum-treated. Then, the samples were observed using a scanning electron microscope (TESCAN MIRA LMS, Brno, Czech Republic) through secondary electron imaging, which was mainly used to determine the presence and distribution of various phases in the sample and to observe the micro-density at the cross-section of the sample.

#### 2.4.5. TG Analysis

After grinding the samples that had been dehydrated and dried into a fine powder that can pass through a 200 mesh sieve, we performed thermogravimetric analysis (TG) testing using a Swiss Mettler Toledo TGA/DSC 3+ instrument (Urdorf, Switzerland). The testing atmosphere was nitrogen, the testing temperature was between 30 and 1000 °C, and the heating rate was 20 °C/min.

#### 2.4.6. Mechanical Properties

The compressive and flexural strength of mortars were tested according to Chinese standards GB17671-2021 [25]. For compressive strength, we used the average of 6 samples, and for flexural strength, we used the average of 3 samples.

## 3. Results

### 3.1. PSD of WCSW

Our size analysis of solid particles in WCSW obtained through different wet-grinding schemes is shown in Figure 1.

The sizes of the WCSW particles corresponding to various wet-grinding treatment schemes are significantly different, and the wet-grinding treatment of scheme 4 has the best effect. The average particle size of the processed WCSW-4 particles is 4.14 μm, and the maximum particle size is only 8.15 μm. Compared with the consolidated CSW, the solid particles in WCSW-4 are smaller, with a narrower particle size distribution range, and the particles are more obviously refined. Therefore, WCSW-4 particles are all used in subsequent tests.

### 3.2. Physical and Chemical Properties of WCSW-4

#### 3.2.1. SEM Analysis

The microstructure images of solid particles in CSW and WCSW-4 observed by SEM are shown in Figure 2. From Figure 2a, there are large-sized solid particles in CSW, and a large number of particles are agglomerated together.

From Figure 2b, the size of solid particles in WCSW-4 is small, and most of them are blocks with relatively flat surface, and because of the small particle size, there are more small particles attached to the surface of large particles. It can be found that wet-grinding treatment significantly reduces the size of solid particles in the CSW and makes the surface of the particles more flat.

#### 3.2.2. XRD Analysis

The XRD images of CSW and WCSW-4 are shown in Figure 3. CSW is obtained by the hydration of cement and fly ash for some time and contains a variety of crystal phases as well as indefinable C-S-H gel. From Figure 3, compared to CSW, in WCSW-4, the diffraction peaks of C_3_S and C_2_S are lower, indicating a lower content; the higher diffraction peak of Ca(OH)_2_ indicates a higher content. Wet-grinding treatment causes more hydration components in CSW to react and generate more hydration products, accelerating the hydration process of minerals.

#### 3.2.3. TG Analysis

The TG curves of CSW and WCSW-4 are shown in Figure 4. After the hydration of cement, there are many phases in the product, which will produce mass changes at different temperatures, and the curve of mass changes to temperature can be obtained by testing. From Figure 4, it can be observed that CSW and WCSW-4 exhibit significant mass changes in several temperature ranges.

The dehydration and decomposition of ettringite and C-S-H gel in the range of 50–250 °C, Ca(OH)_2_ in the range of 400–500 °C and CaCO_3_ in the range of 550–800 °C will cause significant changes in the sample mass [26,27]. Ettringite, C-S-H gel and Ca(OH)_2_ are all products of cement hydration, while CaCO_3_ is formed by the reaction of Ca(OH)_2_ with CO_2_; therefore, the more pronounced the mass decline of the substance during the heating process, the more hydration products it has. WCSW-4 has more significant mass loss in each temperature range, indicating that it has more hydration products and a higher degree of hydration.

### 3.3. Effect of WCSW-4 on Cement Hydration

#### 3.3.1. Hydration Flow

The hydration heat release curves of WCSW-4 particles replacing cement within 72 h are shown in Figure 5.

During the hydration process of cement, a significant exothermic peak is generated due to the hydration reaction and heat release of a large amount of C_3_S minerals [28]. Figure 5a shows that when the amount of cement replaced by WCSW-4 particles increases from 0 to 10%, the time of the hydration heat release peak is gradually advanced from 13.59 to 8.24 h, which is nearly 39.37% earlier, which means that with the increase in the amount of cement replaced by WCSW-4 particles, the hydration of C_3_S minerals in cement will be earlier.

Figure 5b shows that as the amount of WCSW-4 particles replacing cement increases, cement hydration will end the induction period earlier and enter the acceleration stage, and a large number of C_3_S minerals undergo hydration reactions to release heat, accelerating the progress of cement hydration reaction. However, as hydration continues, the difference in accumulated heat release between each group of samples increases first and then decreases, and the difference is not very obvious starting from approximately 60 h. Replacing cement with WCSW-4 particles will advance the heat release of early cement hydration, promoting the progress of early hydration, and with the increase of WCSW-4 particles substitution, there is a more significant promoting effect on cement hydration.

#### 3.3.2. XRD Analysis

The XRD patterns of cement with different amounts of WCSW-4 particles substitution are shown in Figure 6.

Firstly, we conduct qualitative analysis and find that in each group of cement samples, the main hydration product phases include ettringite, Ca(OH)_2_ and CaCO_3_ as well as unhydrated C_3_S and C_2_S components. At different ages, the main phases present in each group of samples remain consistent, and no new phases are generated due to the effect of WCSW-4 particles.

The hydration degree of C_3_S in cement samples obtained through quantitative analysis and calculation is shown in Figure 7.

At 1 d, the hydration degree of C_3_S mineral is the highest in the cement sample with 10% WCSW-4 particle replacement cement content, and then the differences in the hydration degrees of the C_3_S mineral become smaller with the progress of hydration. From 3 d onwards, the hydration degree of C_3_S minerals in each group of cement samples is similar. WCSW-4 particles only promote the early cement hydration and make C_3_S minerals undergo hydration earlier and faster, but this effect is only concentrated in the pre-3 d hydration process and has little effect on the subsequent hydration process.

#### 3.3.3. TG Analysis

The TG curves of cement with WCSW-4 substituted cement are shown in Figure 8. The cement hydration sample will suffer a mass decline during the heating process because of the thermal decomposition of ettringite, C-S-H gel, Ca(OH)_2_, CaCO_3_ and other hydration products at different temperatures. The more significant the mass decline of the sample, the more hydration products it contains. Figure 8a shows that the mass decline of the sample mixed with WCSW-4 particles is more significant at 1 d, which means that more hydration products are thermally decomposed.

WCSW-4 particles replace cement to promote early cement hydration, accelerate the hydration process of C_3_S mineral components within 1 d, and generate more hydration products in the cement sample. From Figure 8b, at 3 d, the mass decline of samples in each group is very similar; that is, the hydration product content of samples in each group is identical, and there is not much difference in the degree of cement hydration.

### 3.4. Effect of WCSW-4 on the Mechanical Properties of Mortar

The strength of mortar at 1 d, 3 d, 7 d and 28 d is shown in Figure 9.

The replacement of cement with WCSW-4 particles enhances the strength of mortar at all ages, and as the amount of WCSW-4 particle substitution increases, the strength of mortar first increases and then decreases. At 1 d, the compressive and flexural strength of the mortar without WCSW-4 addition are 15.41 MPa and 3.93 MPa, and with 7.5 wt.% WCSW-4 particles substitution, these values are 22.14 MPa and 5.71 MPa, increasing by 43.67% and 45.29%. At 28 d, the compressive and flexural strength of the mortar without WCSW-4 addition is 54.43 MPa and 8.25 MPa, and with 5 wt.% WCSW-4 particles substitution, it is 70.36 MPa and 10.06 MPa, increasing by 29.27% and 21.94%. Overall, replacing cement with WCSW-4 particles can effectively enhance the strength, and the effect of strength growth is more significant in the early stage.

## 4. Discussion

The solid particle size in the WCSW obtained after the wet-grinding process is relatively small. When choosing a wet milling treatment scheme with a small grinding medium, high ball grinding speed, and long ball grinding time, the particle size of the WCSW is the smallest, and the effect of wet-grinding treatment is better. Wet grinding is the process of grinding solid materials in a water environment. CSW, water and grinding media are mixed in a ball milling tank, and under the high speed of the ball mill, the movement of the grinding media in the tank is driven, and the large-sized solids in the CSW crack and break apart under the continuous impact and grinding of the grinding media, resulting in smaller particles. At the same time, the presence of a water medium helps dissolve part of the phase; this is conducive to the crushing of large particles and the improvement of grinding efficiency [29,30].

Compared to CSW, WCSW contains more hydration products, which means that wet-grinding treatment not only refines the particle size of solid particles in CSW but also promotes the hydration of cement minerals in it. After the hydration of C_3_S minerals in cement for a period, its surface will gradually be wrapped by the hydration products produced, affecting the dissolution of cement minerals and reducing the hydration rate of cement [31]. In the wet-grinding process, C-S-H gel and other hydration products covered on the surface of C_3_S minerals will be broken by impact, which increases the contact area between C_3_S minerals and water and accelerates the hydration reaction. Therefore, compared with CSW, the hydration products in WCSW are significantly more, and the unhydrated components are reduced considerably.

The results of the hydration heat test indicate that WCSW-4 particles replace cement to promote the early hydration of cement, and the results of XRD and TG analysis also indicate this viewpoint. The early cement hydration mainly involves the dissolution of C_3_S minerals and the formation of C-S-H gel and other hydration products on the surface of cement particles [32], and Figure 10 shows a simulation of the process of early cement hydration. After contact between cement and water, the C_3_S mineral gradually dissolves, and Ca^2+^ is released into the pore solution.

As the Ca^2+^ concentration gradually saturates, nano-C-S-H nuclei begin to form around the C_3_S mineral, and then the nano-C-S-H nuclei continuously promote new C-S-H nucleation and grow [33]. During the nucleation and growth process of C-S-H, Ca^2+^ in the pore solution is consumed, promoting the further dissolution of C_3_S minerals. This process corresponds to the hydration induction and acceleration periods, and the hydration rate is controlled by the nucleation and growth rate of C-S-H [34]. With the progress of hydration, the surface of the C_3_S mineral is gradually covered by the precipitated C-S-H gel and other hydration products, which affects the migration rate of ions, and cement hydration enters the deceleration stage [35]. Due to the influence of the ion migration rate, the hydration rate of cement gradually decreases, and when it reaches a lower stable stage, the hydration process enters a slow stable phase [36].

The promotion mechanism of WCSW-4 particles on early cement hydration is mainly through the nucleation effect. As shown in Figure 10, compared with ordinary cement hydration, in the early stage of hydration, micro-nano-WCSW-4 particles can adsorb Ca^2+^ in the pore solution, which forms an ion concentration gradient and promotes the further dissolution of C_3_S minerals, releasing more Ca^2+^ [28]. The C-S-H gel and CaCO_3_ components in WCSW-4 particles act as excellent nucleation seeds, which can reduce the C-S-H nucleation barrier and promote the rapid nucleation and growth of C-S-H gel on the surface of the WCSW-4 particles [37,38,39,40], and they can also reduce the inhibition of C_3_S minerals dissolution. Therefore, WCSW-4 particles can shorten the hydration induction period, advance the acceleration period, and play a role in promoting the early hydration of cement. Moreover, the higher the replacement amount of WCSW-4 particles, the more additional nucleation sites there are, which can further accelerate the nucleation and growth of C-S-H gel and promote early cement hydration.

Replacing cement with WCSW-4 particles can improve the strength of mortar, especially early strength. After replacing cement with WCSW-4 particles, they can serve as micro–nano-level fillers, making the microstructure and interface transition zone of mortar specimens more dense, resulting in a decrease in porosity and an increase in mortar strength [40,41]. As shown in the SEM images of the mortar sample in Figure 11, most of the mineral components of the sample with a hydration of 28 d have been hydrated, there are more hydration products, the aggregate is well wrapped in it, the overall microstructure is relatively dense, and the mortar has good mechanical properties.

Compared with the ordinary mortar sample, the mortar sample with 5% WCSW-4 substitution has a denser microstructure, smaller and fewer pores, a better combination of hydration products with aggregates, and a denser interfacial transition zone, so it has better mechanical properties. However, the increase of WCSW-4 particle substitution will reduce the amount of cement and enhance the ‘dilution’ effect, which will adversely affect the strength of mortar, so the strength of mortar will decrease when the replacement amount of WCSW-4 particles is high. Micro-nano-scale WCSW-4 particles can serve as nucleation seeds to promote early cement hydration, so they will have a greater improvement in the early strength of mortar.

The wet-grinding process can crush CSW with shorter aging time into smaller particle size WCSW, and at the same time, WCSW can be used as an excellent admixture to reduce cement usage and improve the performance of cement-based materials, providing an effective way for the rapid recovery and utilization of CSW.

## 5. Conclusions

The wet-grinding process can refine CSW. When using zirconia balls with grinding media of 5 mm, 3 mm, and 1 mm, a ball grinding speed of 600 r/min, and a ball grinding time of 1 h, the particle size of WCSW particles is smaller, and the wet-grinding effect is better. Moreover, compared to CSW, the surface of WCSW-4 particles is smoother, with fewer unhydrated components and more hydration products.Replacing cement with WCSW-4 particles can promote the early hydration of cement, and the higher the substitution amount, the more significant the promoting effect. When WCSW-4 particles replace cement, the hydration induction period of cement is shortened, the early hydration of C_3_S minerals is accelerated, and more hydration products are generated.After replacing cement with WCSW-4 particles, the strength of mortar can be significantly improved, and the strength of mortar increases first and then decreases with the increase in WCSW-4 particle substitution amount. At 1 d, the maximum increase in compressive and flexural strength is 43.67% and 45.29%; at 28 d, the maximum increase in compressive and flexural strength is 29.27% and 21.94% with more significant early strength growth.

## Figures and Tables

**Figure 1 materials-17-03010-f001:**
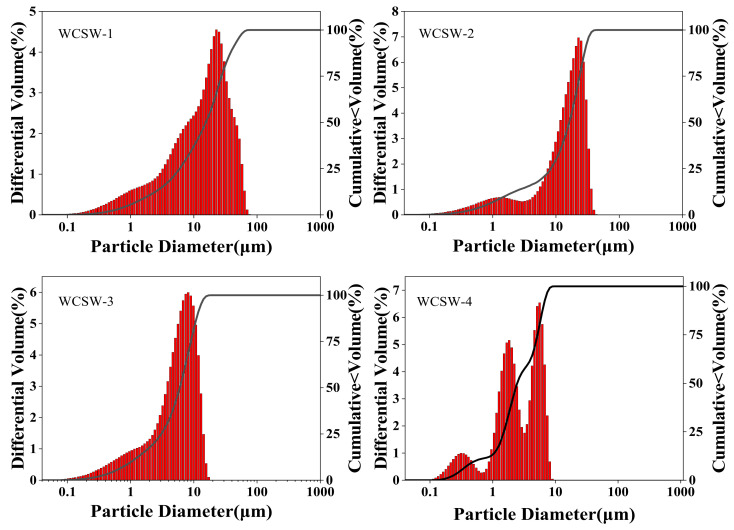
Size analysis of WCSW particles under different wet-grinding schemes.

**Figure 2 materials-17-03010-f002:**
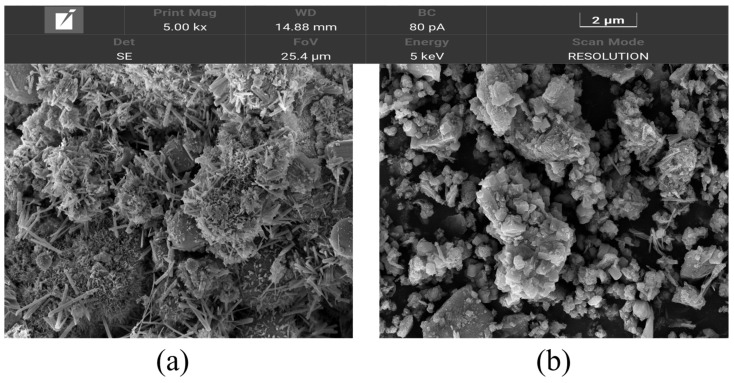
SEM images of (**a**) CSW and (**b**) WCSW-4.

**Figure 3 materials-17-03010-f003:**
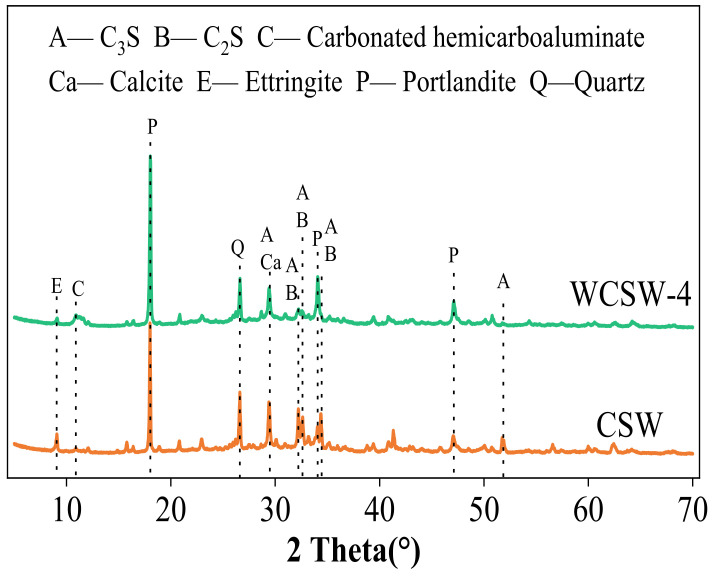
XRD images of CSW and WCSW-4.

**Figure 4 materials-17-03010-f004:**
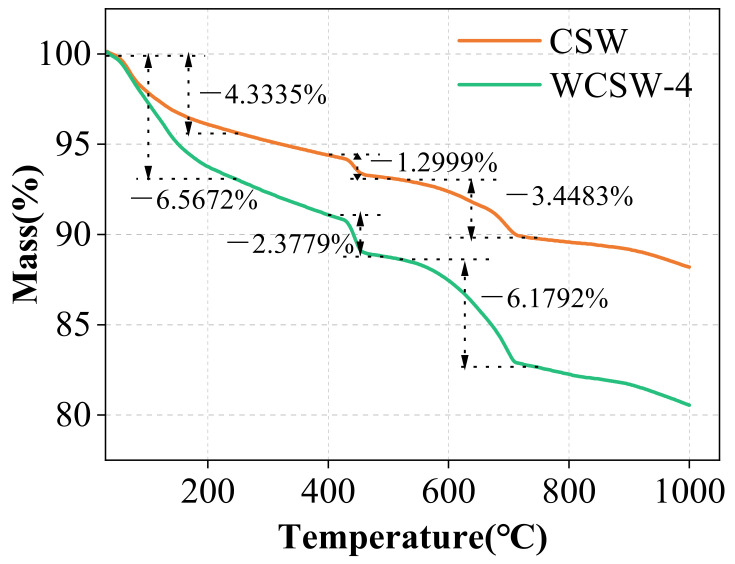
TG curves of CSW and WCSW-4.

**Figure 5 materials-17-03010-f005:**
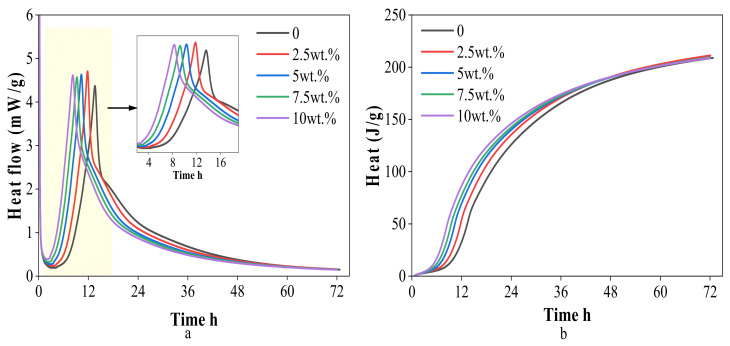
The hydration heat release curves of WCSW-4 particles replacing cement: (**a**) heat flow, (**b**) cumulative heat.

**Figure 6 materials-17-03010-f006:**
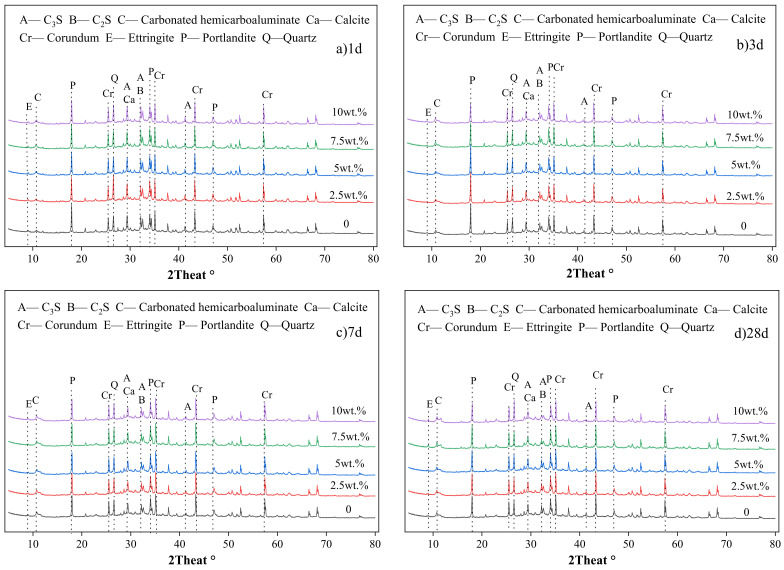
XRD patterns with different WCSW-4 substituted cement.

**Figure 7 materials-17-03010-f007:**
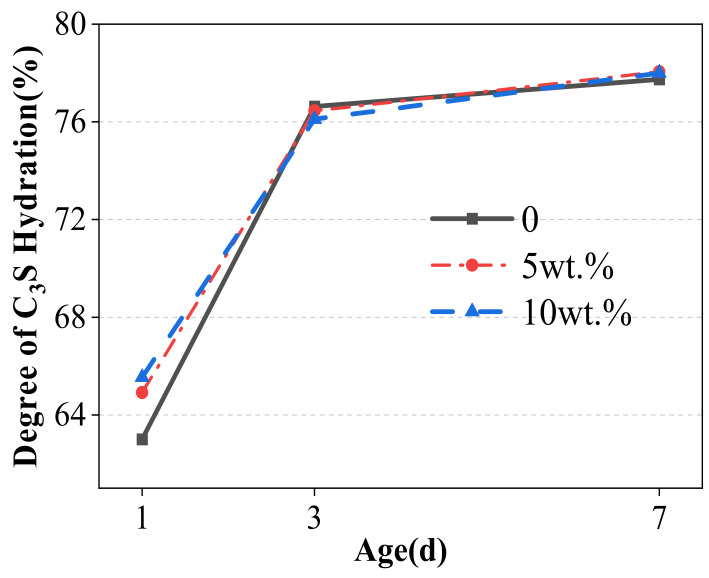
Degree of C_3_S hydration in cement.

**Figure 8 materials-17-03010-f008:**
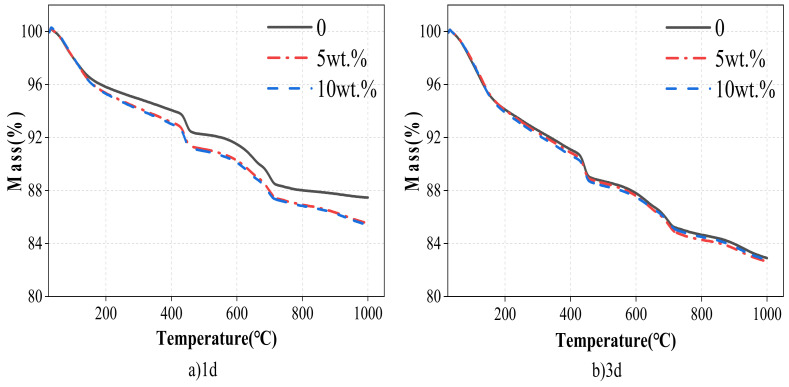
TG curves with different WCSW-4 substituted cement.

**Figure 9 materials-17-03010-f009:**
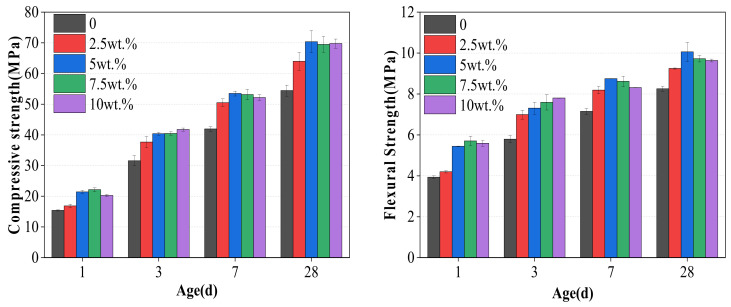
The strength of mortar with different WCSW-4 substituted cement.

**Figure 10 materials-17-03010-f010:**
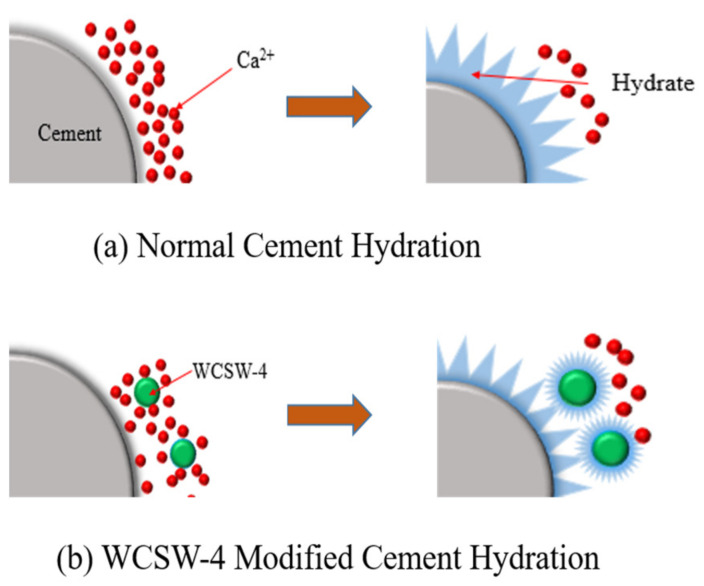
The hydration process of cement with or without WCSW-4.

**Figure 11 materials-17-03010-f011:**
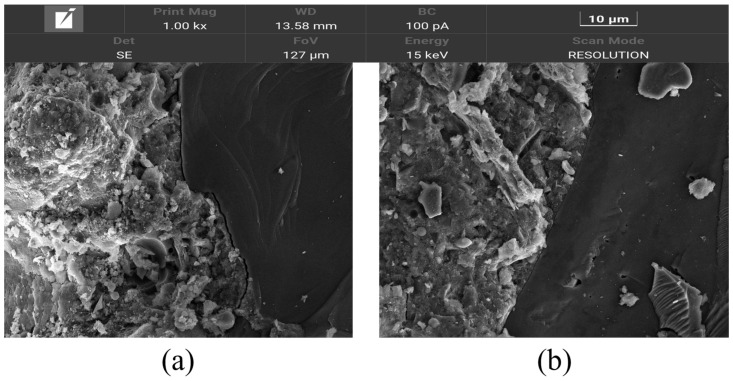
SEM images of (**a**) ordinary mortar and (**b**) mortar with 5% WCSW-4 substitution after hydration for 28 d.

**Table 1 materials-17-03010-t001:** Main chemical components and content of cement and FA.

Components	Cement/%	FA/%
CaO	59.26	4.58
SiO_2_	20.28	44.61
Al_2_O_3_	8.75	33.45
SO_3_	3.94	1.68
K_2_O	0.61	1.09
MgO	2.56	0.35
Fe_2_O_3_	2.41	8.85
Na_2_O	0.78	0.58

**Table 2 materials-17-03010-t002:** The wet-grinding schemes.

Scheme	Grinding Medium	Grinding Speed	Grinding Time
1	15 mm 10 mm 5 mm Agate ball	80 r/min	1 h
2	15 mm 10 mm 5 mm Agate ball	400 r/min	1 h
3	5 mm 3 mm 1 mm Zirconia ball	600 r/min	0.5 h
4	5 mm 3 mm 1 mm Zirconia ball	600 r/min	1 h

**Table 3 materials-17-03010-t003:** Mortar foundation proportions.

Cement/g	FA/g	Sand/g	Water/g	Superplasticizer/g
403.2	76.8	1350	153.6	12

## Data Availability

The data presented in this study are available on request from the corresponding author. The data are not publicly available due to privacy.

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
