# Peer review of "Effect of Wet Grinding Concrete Slurry Waste on Hydration and Hardening Properties of Cement: Micro-Nano-Scale Modification"

_materials, 2024, doi:10.3390/ma17123010_

Round 1

Reviewer 1 Report

Comments and Suggestions for Authors

Comments:

1.       Abstract: Add more quantitative data since this is a research-based study. At the end, add a brief statement on the major implications of this work.

2.       Line 10 and line 13: “Concrete Slurry Waste(CSW)”. Add a space between “(CSW)” and the previous word.

3.       Keywords: Change “strength” to “compressive and flexural strength”.

4.       In the introduction, give the readers about the current state of knowledge in literature by providing the key results and conservations from the previously published studies related to current work. Then, clearly state the knowledge gaps in the past works and their importance to be investigated in this study.

5.       Table 1: Could you mention which materials characterization techniques were used to obtain the data reported in Table 1.

6.       Line 67: “mixed in a ratio of 42:8:200:1.25 to…” On which basis this specific mixed ratio was selected.

7.       The materials and methods section should be reported in past tense, for example, change “is used” to “was used” in line 104; change “samples are observed” to “samples were observed” in line 120, etc.

8.       Figure 10: For better understanding to the readers, in addition to figure, could you also include the cement hydration reactions in the two different cases, i.e., without and with replacement using WCSW.

9.       At the end of discussion, add an individual section/paragraph on the important “Implications and limitations of this study”

10.   This manuscript contains several abbreviations which should be explained at their first appearance in the text. It will improve readability of the manuscript.

Comments on the Quality of English Language

Thorough English language and typographical errors needs to be checked. 

Author Response

  1. Abstract: Add more quantitative data since this is a research-based study. At the end, add a brief statement on the major implications of this work.

Response: Thank you for your comments, we have added related details in line 18 and line 23-25.

  1. Line 10 and line 13: “Concrete Slurry Waste(CSW)”. Add a space between “(CSW)” and the previous word.

Response: Thank you for your comments, we have made the modifications in the manuscript.

  1. Keywords: Change “strength” to “compressive and flexural strength”.

Response: Thank you for your comments, we have made the modifications in the manuscript.

  1. In the introduction, give the readers about the current state of knowledge in literature by providing the key results and conservations from the previously published studies related to current work. Then, clearly state the knowledge gaps in the past works and their importance to be investigated in this study.

Response: Thank you for your comments, we have further supplemented the summary of past research and the important research significance of this study in line 56-59. 

  1. Table 1: Could you mention which materials characterization techniques were used to obtain the data reported in Table 1.

Response: Thank you for your comments, we have added relevant content in line 74.

  1. Line 67: “mixed in a ratio of 42:8:200:1.25 to…” On which basis this specific mixed ratio was selected.

Response: Thank you for your comments, we have added related details in line 77.

  1. The materials and methods section should be reported in past tense, for example, change “isused” to “was used” in line 104; change “samples are observed” to “samples were observed” in line 120, etc.

Response: Thank you for your comments, we have made the modifications in the manuscript. 

  1. Figure 10: For better understanding to the readers, in addition to figure, could you also include the cement hydration reactions in the two different cases, i.e., without and with replacement using WCSW.

Response: Thank you for your comments, in Figure 10, the upper part shows the hydration simulation process of cement without adding WCSW, while the lower part shows the hydration simulation process of cement with adding WCSW.

  1. At the end of discussion, add an individual section/paragraph on the important “Implications and limitations of this study”.

Response: Thank you for your comments, we have added the content of “Implications and limitations of this study” in line 352-355.

  1. This manuscript contains several abbreviations which should be explained at their first appearance in the text. It will improve readability of the manuscript.

Response: Thank you for your comments, we have added relevant content in the manuscript.

Reviewer 2 Report

Comments and Suggestions for Authors

The topic of the research work and manuscript is really interesting and provides new information. However there are some issues to be addressed towards its quality improvement before publication. It would be useful to enrich the description of the state-of-the-art on this specific scientific topic, in order to highlight the gap in knowledge and literature. Additionally, since the number of references used in total in this work is quite low, I would recommend you to adda a brief comment on the effect of aggregates and bio-aggregates potential on the workability of cementitious composites, and  incorporate as well the relevant study of https://doi.org/10.1002/bbb.2291 to strengthen the introduction part and support this comment. You should also strengthen the highlighting of the practical meaning and significance of this work in the final pragraph of the introduction chapter. In materials and methods, could you add as well the number of the specimens used for the determination of each of the properties. For some properties test, you did not refer to a standard used, therefore, please provide the number of standards or relevant previous staudies where you based the process methodology. The DOI numbers are missing in the references list. The whole text of the manuscript is very well-prepared and the figures are attractive and useful. Did you apply any statistical analysis on the results of this work? Since it is not described in materials-methods chapter and I could not find anything in the results as well. It is important you provided at least the standard deviation values.

Comments on the Quality of English Language

Acceptable!

Author Response

Response: Thank you for your comments, according to suggestions, we have further supplemented the research significance in line 56-59, added some testing standards, cited more references, and added the DOI number of the references. But considering the significant difference between the recommended literature and the introduction content, it was not cited. In line 128-131, we have provided a more detailed description of the quantitative analysis of XRD, and in the figure 9, we have provided the standard deviation of compressive and flexural strength of mortar.

Reviewer 3 Report

Comments and Suggestions for Authors

The article is quite interesting and correctly written. I think it would be good to supplement the description of the experiments. Now this is short and the reader must refer to other sources to understand.

I also suggest supplementing the bibliographies with hydrations. Consider working:

Materials, Volume 15, Issue 13July-1 2022 Article number 4403

Author Response

Response: Thank you for your comments, we have added content on experimental details in line 96-98, line 128-131, and line 145-148, and in line 52-55, we have added the content and the references [20-22].

Reviewer 4 Report

Comments and Suggestions for Authors

The paper is good to be published, everything seems well performed, just the experiments like that have been performed earlier in a bit other form. I suggest improve English language and add what is novel in the study and that would be well done

can add some more references on stabilization technologies based on concrete

Comments on the Quality of English Language

grammar and style with English speaking corrector or software

Author Response

Response: Thank you for your comments, we checked the manuscript carefully and revised some mistakes, while added more details.

Round 2

Reviewer 1 Report

Comments and Suggestions for Authors

The revised MS can be accepted. 

Comments on the Quality of English Language

Thorough check of typographical and English language related errors is suggested.